# Preschool Teachers’ Psychological Distress and Work Engagement during COVID-19 Outbreak: The Protective Role of Mindfulness and Emotion Regulation

**DOI:** 10.3390/ijerph19052645

**Published:** 2022-02-24

**Authors:** Mor Keleynikov, Joy Benatov, Rony Berger

**Affiliations:** 1Department of Special Education, Faculty of Education, University of Haifa, Haifa 3498838, Israel; mavrah18@campus.haifa.ac.il; 2Department of Psychology, College of Management and Academic Studies, Haifa 3498838, Israel; 3Bob Shapell School of Social Work, Tel Aviv University, Tel Aviv 6997801, Israel; bergerrony@gmail.com; 4The Center for Compassionate Mindful Education, Tel Aviv 6997801, Israel

**Keywords:** teachers, mindfulness, emotion regulation, COVID-19, work engagement, emotional distress

## Abstract

COVID-19 has dramatically affected the mental health and work environment of the educational sector. Our primary aim was to investigate preschool teachers’ psychological distress and work engagement during the COVID-19 outbreak, while examining the possible protective role of participating in a mindfulness-based intervention geared to foster compassion (Call2Care-Israel for Teachers; C2C-IT) and emotion regulation. The prevalence of emotional distress, work engagement, and COVID-19 concerns were evaluated in 165 preschool teachers in the early stages of the COVID-19 outbreak in Israel through questionnaires. The findings showed that preschool teachers experienced increased emotional distress. Teachers who had participated in the C2C-IT intervention six months before the pandemic outbreak (N = 41) reported lower emotional distress, higher use of adaptive emotion regulation strategies, and higher work engagement, compared to their counterparts that had not participated in the intervention (N = 124). Emotion regulation strategies mediated the link between participating in CTC-IT intervention and emotional distress and work engagement. Teaching is a highly demanding occupation, especially during a pandemic, thus making it important to invest resources in empowering this population. The findings here suggest that the implementation of a mindfulness-based intervention during the school year can enhance teachers’ well-being, even during stressful events such as the COVID-19 pandemic.

## 1. Introduction

Starting in December 2019, coronavirus (COVID-19) spread rapidly and turned into a global pandemic, disrupting the functioning of societies and posing a mental health threat of great magnitude around the world, for both individuals and communities [1]. The fear of the virus and the restrictions imposed due to its outbreak had a tremendous effect on mental health and changed many facets of people’s lives including at work. According to studies, the well-being and engagement of workers in general, and teachers in particular, has been affected [2,3]. Even before the pandemic, teaching identified as a profession that is prone to stress and burnout due to the heavy workload [4]. The COVID-19 outbreak placed new demands on teachers’ already heavy schedules. Teachers have had to cope with students experiencing tremendous social-emotional difficulties, as a result of isolation from their peers, and with overwhelmed parents who struggle to keep their jobs and support their children [5]. Similarly, teachers also needed to equilibrate their personal and professional roles [6]. Another challenge teachers continue to face is the fear of getting infected, and infecting family members [7], as schools are considered a major source of exposure to COVID-19 [8]. These challenges may be even more daunting for teachers working in kindergartens and primary schools, since engaging with preschoolers involves much more physical contact such as hugs, and holding hands, as well as greater hygiene issues [9,10]. During the pandemic, the physical closeness, and the lack of hygiene, typical of preschoolers, constitute a major source of stress for teachers working with this age group [11]. Kindergarten teachers have reported that exposing members of their family or themselves to COVID-19 due to interactions with infected children has been one of the greatest concerns [7].

These stressors have had a detrimental effect on teachers’ mental health [12]. For example, a survey conducted early in the pandemic in over 80,000 teachers in China revealed that the prevalence of anxiety in teachers was almost three times higher than in the general population, with primary school teachers exhibiting the highest level of anxiety [13]. A study of 1633 teachers in Spain working with students of different ages found that kindergarten and primary school teachers manifested the highest anxiety symptoms scores [14]. Two longitudinal studies comparing mental health indicators before and during the pandemic found that during the pandemic, teachers’ anxiety levels were high and that their quality of life and optimism had decreased [2,15]. The difficulties that teachers continue to encounter during the pandemic crisis have affected their work engagement as well [3,16]. Work engagement is an affective-motivational construct defined as a positive, fulfilling, work-related state of mind characterized by high levels of energy while working, strong involvement in one’s work, and concentration and engrossment in work [17]. Work engagement is related to positive outcomes with respect to teachers’ and students’ well-being and work performance [18]. For example, engaged teachers are less prone to burnout [19,20], and their students show higher levels of academic achievement [21]. In addition, there is a strong inverse relationship between teachers’ work engagement and teachers’ drop-out tendencies [19]. Rates of teachers leaving the profession were considered high even before the pandemic, with approximately 23% of new teachers in Israel, and 30–46% of all teachers in the USA leaving the teaching profession before the third year on the job [22,23]. Due to the stress involved in the pandemic, these levels may rise even further [16]. Thus, as a result of the circumstances brought about by the COVID-19 crisis, it is essential to study the factors that may enhance teachers’ ability to cope. A well-documented and timely set of skills for buffering stress, including in educational settings, are mindfulness practices [24,25], yet the underlying mechanism for this buffering effect remains unclear. To shed light on the mechanisms involved in the relationship between mindfulness and emotional distress, the present study implemented the Coffey et al. [26] model. Using factor analysis and structural equation modeling, the authors found that mindfulness, through clarity about one’s own experience, improves the ability to deal with negative emotions. That is, the Coffey et al. model posits that emotion regulation underlies the salutary effects of mindfulness on well-being. Based on that notion, the present study examined whether emotion regulation is a potential mediator of the beneficial link between mindfulness and a compassion-based intervention and emotional distress and work engagement in preschool teachers during the early stage of the COVID-19 outbreak.

### 1.1. Literature Review

#### 1.1.1. Mindfulness

Mindfulness refers to the capacity to devote one’s complete attention to the experiences occurring in the present moment, in a nonjudgmental accepting way [27]. There is growing empirical evidence supporting the existence of a link between mindfulness and well-being, and a negative association between mindfulness and mental health issues, such as depression, anxiety, and stress [28]. In recent years, mindfulness has begun to receive attention in the organizational psychological literature [29,30,31], since mindfulness reported to have a positive relationship with work performance, work engagement, and employees’ well-being [30,31,32,33,34]. According to [34] mindfulness modulates employees’ dysphoric mood states through the cultivation of compassion and self-compassion, improves the ability to control stress, the use of more effective coping strategies, and the adoption of an accepting and non-judgmental attentional set. Mindfulness is especially important for teachers since teaching is an exceedingly challenging and stressful profession [35,36]. Mindfulness-based interventions (MBIs) have been shown to have positive effects on teachers’ mental health including the enhancement of well-being, self-compassion, emotional awareness in the classroom, and reduction in stress, burnout, and depression (for a review see [37]). A prospective controlled study on teachers during the first COVID-19 lockdown showed an MBI intervention enhanced resilience and improved well-being among teachers, while the control group exhibited an increase in burnout levels and a decline in psychological well-being [38]. In addition, mindfulness skills were shown to impact teachers’ emotional support in class and relationships with children, by enhancing self-compassion and compassion for others, and by increasing empathy, social connectedness, and emotional intelligence [36,39].

Contemporary mindfulness interventions often include components of compassion. Compassion defined as sensitivity to suffering in the self and others, with a commitment to try to alleviate it [40]. Some theoretical frameworks associate mindfulness and compassion. For instance, the Mahayana Buddhist tradition posits that mindfulness serves to cultivate awareness for the suffering of the self and others, whereas compassion is the motivational drive that eventually leads to caring behaviors and well-being. Another theoretical framework that underpins the connection between these concepts is Gilbert’s [41] social mentality theory, which suggests that the practice of mindfulness sets the stage to engage with suffering, but that the development of compassionate competencies is necessary to foster growth in the self and others. Compassion considered a pivotal aspect of mindfulness practice since learning how to be forgiving with oneself and others is a key component in mindfulness practice.

#### 1.1.2. Emotion Regulation

Several mechanisms are considered to underlie the salutary effects of mindfulness, a mechanism that has consistently proposed as a central process is emotion regulation (for a review see [42]). Emotion regulation (ER) defined as a cognitive way of managing emotionally arousing information [43]. The process of ER includes different ER strategies that can broadly be divided into adaptive and maladaptive strategies [43,44]. In this study we chose to focus on four common ER strategies in research and therapy: reappraisal, acceptance, rumination, and catastrophizing. Reappraisal and acceptance are typically considered adaptive strategies, in that they usually downregulate negative emotions. On the other hand, rumination and catastrophizing considered maladaptive regulation strategies because they may prolong or even deepen negative emotions [43]. Although yet to be tested during a crisis, studies conducted during routine periods of time have found that teachers who tend to use adaptive ER have lower levels of psychological symptoms, beyond what was predicted by work-related stressors and showed lower burnout and higher engagement in their occupational role [6,45,46,47]. Therefore, enhanced ER is likely to lead to positive outcomes for teachers during COVID-19.

Mindfulness can also lead to more adaptive ER in a variety of ways. Generally speaking, mindfulness triggers processes that optimize psychological functioning, such as the ability to manage emotions [26]. This is because compassionate awareness may alter individuals’ relationships to their own internal experiences in ways that may directly reduce the intensity of their emotional responses [48]. Further, mindfulness skills enhance the use of adaptive regulation strategies such as reappraisal and acceptance and reduce the maladaptive use of rumination and catastrophizing. For example, non-judgmental awareness, which is a key component in mindfulness [27], facilitates healthy involvement in emotions, thus allowing people to experience and express their emotions without trying to reject but rather accept them [49]. Consequently, practicing mindfulness may lead to higher use of acceptance. Likewise, mindfulness is a state in which one attends to the present rather than worry about the future or be concerned about the past, leading to substantially reduced use of catastrophizing and rumination. Finally, mindfulness is a metacognitive form of awareness that involves shifting cognitive sets that make it possible to evaluate life events in an alternative way, leading to higher reappraisal and lower catastrophizing [50]. Therefore, ER strategies may serve as a mediator to explain the effectiveness of MBIs on psychological distress and work engagement.

#### 1.1.3. The Teaching Context

In recent years, MBIs have been adapted and applied in the teaching context [37]. For teachers, practicing mindfulness meditation may be an effective means of reducing stress and burnout while enhancing well-being via the promotion of ER and compassion abilities [37,51]. For instance, Mindfulness-Based Stress Reduction for teachers (MBSR) [52] is a mindfulness-training program based on meditation techniques that seek to change teachers’ relationships with stressful events. Theoretically, MBSR may reduce symptoms of stress by modifying self-compassion, ER processes, and improving teachers’ ability or willingness to allow and accept negative emotions rather than attempt to suppress or avoid them [49]. This theory confirmed in studies on the general population [53], clinical populations [54], and teachers [55] but never been tested during a major crisis.

The Call2Care-Israel for Teachers (C2C-IT) is a mindfulness and compassion-based intervention geared to foster self and other compassion/care [56]. This intervention accredited by the Ministry of Education and offered as in-service training for all teachers, as part of the Ministry’s efforts to impart mental resilience to teachers. In the intervention, three modes of care: receiving care, developing self-care, and extending care, under the assumption that each mode empowers the others, are structured in a sequence that includes mindfulness meditations, social skill training, and group activities. Preliminary results indicated that the C2C-IT intervention had a significant effect on reducing teachers’ stress levels and promoting their well-being [57]. Testing whether ER strategies mediate the effects of the C2C-IT intervention and its efficacy in the context of a pandemic may be imperative. Although the effectiveness of mindfulness and compassion-based interventions for the well-being of teachers extensively tested, few studies have examined the mechanisms mediating the effects of such interventions. In addition, assessing whether such interventions contribute to resilience under acute stress conditions, such as COVID-19 outbreaks, is crucial for determining their effectiveness at times when it is most required.

### 1.2. The Current Study

The main objective of the present study was to investigate preschool teachers’ psychological distress and work engagement during the early stages of the COVID-19 outbreak, while examining the possible protective role of participating in the C2C-IT intervention and the mediating role of ER strategies. To the best of our knowledge, no study has examined the relationship between participating in a mindfulness-compassion-based intervention and emotional distress and work engagement in times of crisis. Hence, the research aims were as follows: (1) to evaluate the extent of preschool teachers’ exposure to the COVID-19 pandemic during the first quarantine in Israel (May 2020) and to evaluate the extent to which preschool teachers experienced COVID-19 related concerns, emotional distress, work engagement during the first quarantine in Israel. (2) To explore the differences between teachers who participated in the C2C-IT intervention before the COVID-19 outbreak and teachers who did not participate in the intervention, on measures of emotional distress, work engagement and ER tendencies during the pandemic. (3) To examine whether ER strategies mediated the association between participation in the CTC-IT intervention and emotional distress and work engagement. The hypotheses were as follows. (1) Preschool teachers will report increased emotional distress symptoms and reduced work engagement. (2) Preschool teachers who participate in the intervention will report higher work engagement, enhanced use of ER strategies, and fewer symptoms of emotional distress, compared to teachers who did not participate in the intervention program. (3) Based on the Coffey et al. [26] model, ER strategies will mediate the link between participating in C2C-IT intervention and emotional distress and work engagement.

## 2. Method

### 2.1. Recruitment and Study Procedure

Cross-sectional data collected online through Google forms, during April–May 2020. At the time, Israel experienced a surge in the number of daily new cases. The ongoing rise in the numbers of diagnosed cases affected the public’s estimations of the severity and controllability of the virus [58] and led to emergency regulations. These regulations, which issued to curb the spread of COVID-19 in Israel by minimizing social contact, included restrictions on movement and gatherings. As of 12 March, all early educational frameworks closed, and discussions about when and under which conditions to reopen the education system held within the government and Ministry of Education. Finally, the early educational institutes reopened on 10 May. The questionnaires distributed to the Ministry of Education’s kindergarten teachers through the Ministry’s supervisors. This study approved by the ethics committee of the chief scientist of the Ministry of Education. Participation in this study was voluntary and anonymous. The C2C-IT intervention is one of the social-emotional training interventions the Ministry of Education offers to teachers as part of their professional development. Teachers can choose to attend this intervention for credit or for qualifications. Participants in the C2C-IT group chose to take part in the intervention as part of credit toward career education offered by the Teaching Development Center of the Ministry of Education. All kindergarten teachers in Israel could participate. The inclusion criteria for participants in the current study was being a female preschool teacher. In total, there were 20 respondents who taught in primary schools; hence, their data were excluded from the study.

### 2.2. Intervention

The C2C-IT intervention first administered in May–June 2019, 6 months before the pandemic. The intervention is composed of 20 sessions that address three modes of care: receiving care, self-care, and extending care (see Appendix A for details). Each session in the three modes includes modules on psychoeducational material (e.g., mindfulness and compassion effects on brain activity, correlates of mindfulness and compassion), contemplative practices (e.g., teaching mindful breathing, body scan, or caring-figure meditation), social-emotional skills (e.g., identifying and sharing emotions, learning to receive and give social support, and empathy skills), and group activities (e.g., sharing feelings with peers or role-playing). These accompanied by homework assignments (e.g., practicing compassion, paying attention to automatic reactions in challenging situations, or body scan) and presented in the Appendix A. A previous study showed that the intervention to increased well-being in teachers [57].

### 2.3. Instruments

The survey questionnaire used in this study included items on participants’ demographics, COVID-19 level of exposure, COVID-19 related concerns, ER strategies, work engagement, and psychological distress.

*Demographic questionnaire.* The demographic questionnaire aimed at collecting background variables such as age, gender, family status, and education. We also included questions concerning direct exposure to COVID-19 (e.g., “Are you in quarantine due to an exposure to individuals with COVID-19?”; “Has one of the most important people in your life been diagnosed with COVID-19?”). We further asked participants to indicate the last five digits of their ID, for purposes of cross-referencing with participants in the C2C-IT intervention.

*COVID-19 concerns questionnaire.* Aimed at assessing the level of concerns arising from the COVID-19 pandemic (e.g., “I am worried about my future financial situation”). Participants responded to this questionnaire on a 5-point scale ranging from 1 (not at all) to 5 (very much). This questionnaire was based on a measure of teachers’ stress about the COVID-19 outbreak administered by [59] in a study conducted in Hebrew.

*Cognitive Emotion Regulation Questionnaire-short form* (CERQ-SF) [43]. The CERQ-SF is an 18-item self-report measure of nine cognitive ER strategies (self-blame, acceptance, rumination, positive refocusing, refocus on planning, positive reappraisal, putting into perspective, catastrophizing, and other-blame), each of which involves two items. Participants are asked to rate how they cope with negative events on a 5-point Likert-type scale ranging from 1 (rarely) to 5 (almost always). A high score reflects the primary use of a CERQ strategy. The Cronbach alphas for the different strategies ranged from 0.61 to 0.90 in the current study.

*Utrecht Work Engagement Scale* (UWES) [60]. The UWES is a nine-item questionnaire that assesses levels of work engagement. The scale consists of three subscales: absorption (e.g., “I am immersed in my work”), vigor (e.g., “At my job I feel strong and vigorous”), and dedication (e.g., “My job inspires me”). High scores on all three dimensions indicate high work engagement. Items are scored on a scale ranging from (0) “never” to (6) “always”. The Cronbach alpha for the UWES overall scale was 0.92 in the current study.

*Depression, anxiety, and stress scale* (DASS-21) [61]. The DASS-21 used to assess psychological problems. This 21-item self-report scale has three different categories: Depression, Stress, and Anxiety consisting of seven items for each domain. Participants are asked to rate the frequency of experiencing negative emotions over the previous week on a 4-point Likert scale (0 = did not apply to me at all; 3 = applied to me very much, or most of the time). The Cronbach alpha for the DASS domains assessed in the current study was 0.89 for depression, 0.88 for anxiety, 0.90 for stress, and 0.96 for the overall scale.

### 2.4. Analysis

Descriptive statistics calculated to get a clearer picture of the extent of exposure to COVID-19, COVID-19 concerns, emotional distress, and work engagement among preschool teachers in Israel during the first quarantine. To examine the unique contribution of participation in the C2C-IT intervention to emotional distress, independent T-tests were conducted comparing preschool teachers who participated in the C2C-IT intervention vs. the control groups. All the analyses conducted using SPSS.24 software (Armonk, NY: IBM Corp). Since the [26] theoretical model suggests that mindfulness may have a positive effect on emotional distress and work engagement by improving emotion regulation abilities, we employed the PROCESS mediation macro in SPSS (Model 4; [62]) to estimate the indirect effects of C2C-IT intervention on emotional distress and work engagement via ER strategies as the mediators. The specific indirect effects of the independent variable on the dependent variable through a mediator are the product of two paths (a, b) linking the independent variable to the dependent variable via a mediator. When the confidence intervals of the indirect effect of a mediator do not include 0, it is considered statistically significant. Mediators tested by calculating the bias-corrected 95% CIs using bootstrapping with 5000 resamples via the Process procedure for SPSS. Demographic variables in which the groups differed significantly (i.e., age, seniority, and marital status) were included as control variables for analysis, to examine the possible effects of these factors on the observed mean difference.

## 3. Results

### 3.1. Demographic Characteristics

The current study was composed of 165 female preschool teachers (mean age = 43.3 years, age range = 25–64, SD = 9.03), including 41 teachers that took part in the C2C-IT group and 124 teachers in the control group. The participants had an average of 15.9 years of seniority, and most of them work in general education (79%). In terms of the level of religiosity, 46% of the participants were secular, 27% were traditional, 24% were religious and 3% were Ultra-Orthodox. As shown in Table 1, comparing the C2C-IT group to the control group showed no differences in terms of the participants’ number of children, education level, education system, religiosity, or COVID-19 level of exposure. By contrast, the group differed in terms of age, years of seniority, and marital status. Specifically, the participants in the C2C-IT group were significantly younger, had fewer years of seniority, and differed in their marital status. The intervention group was on average slightly younger and had less work experience compared to the control group.

### 3.2. Group Differences in the Study Variables

To evaluate preschool teachers’ exposure to the COVID-19 pandemic, levels of COVID-19 related concerns, ER tendencies, emotional distress, and work engagement, as well as to explore the differences between teachers who participated in the C2C-IT intervention before the COVID-19 outbreak and teachers who did not participate in the intervention on these measures, independent sample T-tests and chi-square analyses conducted (Table 2). The three top concerns rated by preschool teachers as the most worrying were concerns about exposing themselves or members of their family to COVID-19, concerns regarding their family’s economic status, and concerns about a decline in the emotional functioning of the children in kindergarten. The C2C-IT group reported lower COVID-19 concerns [*t*(163) = 7.47, *p* < 0.01] than the control group. The C2C-IT group, as compared to the control group, reported lower emotional distress [*t*(163) = 9.83, *p* < 0.01], and higher levels of work engagement [*t*(163) = −8.27, *p* < 0.01]. The groups also differed in their habitual use of all emotion regulation strategies. Specifically, the C2C-IT group used more reappraisal [*t*(163) = −2.54, *p* < 0.01] and acceptance [*t*(163) = −6.73, *p* < 0.01], and engaged to a lesser extent in rumination [*t*(163) = 3.73, *p* < 0.01] and catastrophizing [*t*(163) = 6.31, *p* < 0.01] compared to the control group.

### 3.3. Mediation Models

To determine whether ER strategies mediated the relationship between participation in the C2C-IT course and emotional distress, mediation models were conducted for each strategy separately (see Figure 1a). Because the participants of the C2C-IT group differed from the participants in the control group in terms of age, marital status, and seniority, these variables entered as covariates. The total effect of participating in the C2C-IT intervention on emotional distress (i.e., the effect of participating in the C2C-IT intervention on emotional distress while not controlling for the mediators) was statistically significant. In addition, the direct effect of participating in the C2C-IT intervention on emotional distress (i.e., the effect of the independent variable on the dependent variable while controlling for the mediator) was significant for all the ER strategies. The mediation analysis revealed an indirect effect (i.e., the effect of participating in the C2C-IT intervention on emotional distress through a particular mediator) for reappraisal (indirect effect = 2.72, SE = 0.86, *p* < 0.05, 95% CI [−3.55, −0.20]), catastrophizing (indirect effect = −3.13, SE = 1.11, *p* < 0.01, 95% CI [−5.67, −1.23]). However, the indirect effect of group type and emotional distress via acceptance (indirect effect = −0.21, SE = 1.14, *p* = 0.42 95% CI [−2.49, 2.03]) and rumination (indirect effect = −0.69, SE = 0.64, *p* = 0.13, 95% CI [−2.13, 0.48]) was not significant; see Table 3.

Mediation models conducted for each strategy separately to test the indirect effects of the C2C-IT intervention on work engagement through ER strategy (see Figure 1b). The total effect of participating in the C2C-IT intervention on work engagement was statistically significant. In addition, the direct effect of participating in the C2C-IT intervention on work engagement was significant for all four ER strategies examined. These results thus suggested that the three emotion regulation strategies mediated this relationship (see Table 3). We found significant indirect effects for reappraisal (indirect effect = 0.23, SE = 0.16, *p* < 0.05, 95% CI [−0.04, 0.44]), catastrophizing (indirect effect = −0.28, SE = 0.11, *p* < 0.01, 95% CI [0.10, 0.51]), and acceptance (indirect effect = −0.41, SE = 0.13, *p* < 0.01, 95% CI [0.20, 0.71]) whereas the mediation effect of rumination was not significant (indirect effect = −0.01, SE = 0.06, *p* = 0.46, 95% CI [−0.13, 10]).

## 4. Discussion

The present study had three aims including: (1) Investigating preschool teachers’ psychological distress and work engagement during the early stages of the COVID-19 outbreak; (2) Explore the possible protective role of participating in the C2C-IT intervention on preschool teachers’ mental health and work engagement; (3) Examine whether ER strategies mediate the relationship between participation in the C2C-IT intervention and mental distress and work engagement. The findings suggest that the intervention group used more effective ER strategies, had fewer symptoms of mental distress, and reported on higher work engagement compared to the control group. In addition, reappraisal and catastrophizing found to mediate the relationship between participation in the CTC-IT intervention and mental distress and work engagement. Findings discussed below and should be examined further in future research.

Regarding the first aim of the current study, we found that during the first lockdown in Israel, there was low actual exposure to the virus among teachers, but great uncertainty and concern. The most worrying concern reported by the preschool teachers had to do with becoming infected or infecting family members. This finding is consistent with other studies that have found that fear of infection is one of the most significant causes of anxiety among teachers, which probably stems from teachers’ close interaction with children daily [7,63]. The stress, anxiety, and depression symptomatology rates of preschool teachers in the current study were somewhat high in comparison to those reported in other studies conducted during that time in the general population in Israel [64,65]. Specifically, whereas the mean total score on the DASS among teachers was 12.7, in other studies the mean total score ranged from 10.72 to 12.03 in the general population. This suggests that in workplaces that involve physical contacts such as schools, hospitals, and physiotherapy clinics, employees experience high emotional distress during the pandemic [66,67,68,69].

Studies have shown that mindfulness and compassion-based interventions have positive outcomes on teachers’ mental health during routine periods (for a review see [36]) and during the COVID-19 outbreak [38]. More specifically, teachers who participated in mindfulness and cognitive reframing intervention programs reported less mental distress and burnout, despite the COVID-19 pandemic, as compared to controls. A recent study conducted in Italy found that an MBSR intervention can reduce the negative psychological implications of the COVID-19 pandemic on teachers’ wellbeing during a lockdown. Teachers who participated in the C2C-IT intervention reported lower mental distress and a higher sense of efficacy. However, the long-lasting associations between participating in MBI interventions and emotional distress and work engagement in times of crisis remain unknown. To contribute to clarifying this relationship, the current study examined whether those preschool teachers who participated in the intervention would report higher work engagement, enhanced use of ER strategies, and fewer symptoms of emotional distress compared to teachers who did not participate in the intervention. The findings confirm this hypothesis since a positive association between the C2C-IT intervention and distress during a crisis was found. The results indicate that teachers who participated in the C2C-IT intervention 6 months before the pandemic reported improved ER skills, higher work engagement, and lower emotional distress. These results are highly relevant in Israel, as the Israeli population is subject to additional stressors originating from political violence. Large classes also constitute a stressor since Israel is a country with one of the highest classroom densities in the world, with approximately 26.5 students per class on average and a maximum of 34 students per class from the age of 3, compared to an average of 21.1 students in other OECD countries [70].

Many interventions aim to enhance teachers’ coping abilities with mental distress (for a review see [71,72]), yet most have not examined the mechanisms through which the intervention achieves its goal. ER has been described as a mechanism of change in mindfulness and compassion-based interventions [26,42]. It has suggested that mindfulness may facilitate a more positive and accepting stance towards emotions as opposed to becoming overwhelmed or ruminating over these experiences [49]. Accordingly, the findings here suggested that teachers who participated in the C2C-IT intervention tended to use more adaptive ER strategies, such as acceptance and reappraisal, and showed a lower tendency to use strategies that considered maladaptive, such as rumination and catastrophizing. That is, the C2C-IT appeared to have lessened over-engagement with distressing thoughts and emotions (e.g., lower catastrophizing and rumination) and facilitated positive and non-judgmental appraisals of experience (e.g., higher reappraisal and acceptance).

The third hypothesis based on the Coffey et al. model [26], which posits that ER may be the mechanism through which mindfulness affects mental distress. Consistent with this notion, we hypothesized that ER strategies would mediate the association between participation in the C2C-IT intervention and symptoms of mental distress and work engagement among pre-school teachers. This hypothesis partially confirmed. That is, indirect effects of the intervention on emotional distress and work engagement through reappraisal and catastrophizing were found. Hence participation in the C2C-IT intervention may have led to an increase in positive appraisals (i.e., reappraisal), and reduced negative assessments (i.e., catastrophizing), thus leading to reduced symptoms of emotional distress and higher work engagement. Thus, the salutary effects of mindfulness on emotional distress and work engagement may be mediated by strengthening a positive cognitive-emotional process and by disrupting a negative one. However, the indirect effect of group type and emotional distress via acceptance and rumination was not significant. It is possible that in crises, repetitive thinking in the form of rumination is somewhat natural and not necessarily an indicator of psychopathology.

Taken together, the findings point to the positive, long-lasting effects of a C2C-IT intervention on emotional distress and work engagement of preschool teachers, which maintained even in the high-intensity negative emotional situation of the COVID-19 outbreak. The current findings suggest that through the practice of mindfulness, individuals may be able to develop an expanded, nonjudgmental state of present-moment awareness that facilitates positive reappraisal and acceptance of stressful life events while reducing a tendency to catastrophize these events, which may finally lead to reduced emotional distress and increased work engagement.

### Limitations

This study has a few limitations that should be acknowledged. First, it consisted of a cross-sectional exploration of post-intervention implications. Although this method has been applied in other studies [73,74], it is not the classic before and after design, so causality cannot be inferred, and baseline levels are unknown. Nevertheless, the groups were reasonably well matched in terms of their demographic characteristics. Some of the differences (e.g., age, seniority, and marital status) were included in the analysis as control variables. Future studies should measure the variables before beginning the intervention program so that causality can be inferred. Furthermore, we assessed all constructs via self-report, which are susceptible to socially desirable responses, potentially yielding biased results. Future research would benefit from examining ER, work engagement, and emotional distress using other types of assessment (e.g., behavioral, or physiological measurements, interviews, or experimental manipulation). Further, the sample size was relatively small and composed only of females, limiting the generalizability of the results. Finally, we collected the data in the early stages of the COVID-19 outbreak in Israel. Consequently, the period of exposure to the pandemic was short, such that the findings might not be generalizable to long-term emotional implications.

## 5. Conclusions

Teachers’ work environment, which includes physical contact with many children, little opportunity to maintain hygiene and wear masks as recommended, may result in psychological distress, anxiety, and depression. Therefore, it is of utmost importance to find protective factors to shield against the mental distress of this population. The findings here showed that teachers who participated in a mindfulness and compassion intervention six months before the pandemic outbreak showed lower symptomology rates of emotional distress during the COVID-19 outbreak compared to teachers who did not participate in the intervention. This study thus provides preliminary evidence that participating in a C2C-IT intervention may help teachers sustain psychological wellbeing, even during a major crisis. The results suggest that the associations between mindfulness and compassion intervention and psychological distress and work engagement mediated by applying more adaptive ER strategies, thus suggesting that ER is one of the mechanisms underlying the deployment of teachers’ mindfulness. The COVID-19 pandemic is entering its third year; hence, mental difficulties may accumulate and the commitment to work may decrease, especially among preschool teachers. Reality cannot be controlled, but the interpretation of reality, and consequently the effect on the state of mind is to some extent something that can be practiced. Therefore, the Ministry of Education and other labor organizations should consider mindfulness-based interventions as a practical approach to improving well-being and work engagement, even during stressful events such as the COVID-19 pandemic. In addition, the relevant bodies should allocate resources to the rehabilitation of teaching staff at the end of the crisis, as they were found to be experiencing high emotional distress. From a scientific point of view, longitudinal studies are required to examine which intervention programs and protective factors maintain their effectiveness over time. Future studies can build on this work by examining the longitudinal role of mindfulness and compassion-based interventions on other related variables, such as burnout and resilience.

## Figures and Tables

**Figure 1 ijerph-19-02645-f001:**
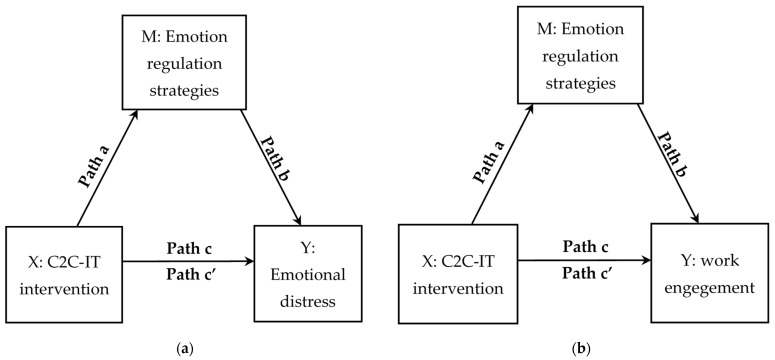
Conceptual model of the mediation model. (**a**) Emotion regulation strategies as a mediator be-tween the C2C-IT intervention and emotional distress. (**b**) Emotion regulation strategies as a mediator between the C2C-IT intervention and work engagement.

**Table 1 ijerph-19-02645-t001:** Demographic characteristics of participants in the study (N = 165).

	C2C-IT Group (*n* = 41)	Control Group (*n* = 124)	Total (*n* = 165)	Statistics
Age [M (SD)]	40.2 (8.79)	44.3 (8.91)	43.3 (9.03)	*t*(163) = 2.57 **
Number of children [M (SD)]	2.5 (1.38)	3.0 (2.39)	2.9 (2.18)	*t*(163) = 1.24
Years of seniority [M (SD)]	12.8 (7.85)	16.9 (8.92)	15.9 (8.82)	*t*(163) = 2.62 **
Education (%)				
High school	3 (7%)	2 (2%)	5 (3%)	*χ*^2^(2) = 4.19
Bachelor’s degree	27 (66%)	77 (62%)	104 (63%)	
Master’s degree or higher	11 (27%)	45 (36%)	56 (34%)	
Marital status (%)				
Single	5 (12%)	10 (8%)	15 (9%)	*χ*^2^(4) = 10.88 *
In a relationship	9 (22%)	7 (6%)	16 (10%)	
Married	26 (63%)	101 (82%)	127 (77%)	
Divorced	1 (2%)	5 (4%)	6 (4%)	
Widow	0 (0%)	1 (1%)	1 (1%)	
Education system (%)				
Special Education	1 (2%)	5 (4%)	6 (4%)	*χ*^2^(3) = 4.09
State Education	5 (12%)	23 (19%)	28 (17%)	
Religious Education	1 (2%)	0 (0%)	1 (1%)	
General Education	34 (83%)	96 (77%)	130 (79%)	
Religiosity (%)				
Secular	25 (61%)	51 (41%)	76 (46%)	*χ*^2^(3) = 6.29
Traditional	7 (17%)	37 (30%)	44 (27%)	
Religious	7 (17%)	33 (27%)	40 (24%)	
Orthodox	2 (5%)	3 (2%)	5 (3%)	

Note: *p* < 0.05 *, *p* < 0.01 **.

**Table 2 ijerph-19-02645-t002:** Group differences in the research measures.

	C2C-IT Group (*n* = 41)	Control Group (*n* = 124)	Total (N = 165)	Statistics *t*(163)
COVID-19 exposure [M (SD)]	0.1 (0.3)	0.2 (0.5)	0.1 (0.4)	0.39
COVID-19 concerns [M (SD)]	29.2 (6.9)	40.2 (11.4)	37.5 (11.4)	7.47 **
Acceptance [M (SD)]	4.6 (0.9)	3.4 (1.0)	3.7 (1.1)	−6.73 **
Reappraisal [M (SD)]	4.5 (0.6)	4.2 (0.8)	4.3 (0.8)	−2.54 *
Catastrophizing [M (SD)]	1.3 (0.7)	2.2 (1.0)	2.0 (1.0)	6.31 **
Rumination [M (SD)]	3.0 (0.9)	3.6 (0.9)	3.5 (0.9)	3.73 **
Emotional distress [M (SD)]	3.0 (3.4)	15.9 (13.4)	12.7 (13.0)	9.83 **
Work engagement [M (SD)]	6.6 (0.4)	5.6 (1.2)	5.9 (1.1)	−8.27 **
Stress [*n* (%)]				*χ*^2^(2) = 23.5 **
Normal	40 (98%)	70 (57%)	110 (67%)	
Mild-moderate	1 (2%)	26 (21%)	27 (16%)	
Severe—extremely severe	0 (0%)	28 (23%)	28 (17%)	
Anxiety [*n* (%)]				*χ*^2^(2) = 19.0 **
Normal	40 (98%)	77 (62%)	117 (71%)	
Mild-moderate	1 (2%)	17 (14%)	18 (11%)	
Severe—extremely severe	0 (0%)	30 (24%)	30 (18%)	
Depression [*n* (%)]				*χ*^2^(2) = 19.1 **
Normal	38 (93%)	69 (56%)	107 (65%)	
Mild-moderate	3 (7%)	31 (25%)	34 (21%)	
Severe—extremely severe	0 (0%)	24 (19%)	24 (15%)	

Note: *p* < 0.05 *, *p* < 0.01 **.

**Table 3 ijerph-19-02645-t003:** Mediation of the effect of the C2C-IT intervention (IV) on the dependent variables (DV) through ER strategies (N = 165).

Mediating Variable (M)	Effect of IV on M (a)	Effect of M on DV (b)	Direct Effects (c′)	Indirect Effect (ab)	95% CI		R^2^	F
					Boot LLCI	Boot ULCI		
**Emotional distress (DV), Total effect (c) = −12.42 ****								
Reappraisal	0.29 *	−5.31 **	−10.87 **	−1.56 *	−3.55	−0.20	0.29	12.79 **
Acceptance	1.17 **	−0.18	−12.21 **	−0.21	−2.49	2.03	0.19	7.54 **
Catastrophizing	−0.81 **	3.90 **	−9.30 **	−3.13 **	−5.67	−1.23	0.27	11.74 **
Rumination	−0.55 **	1.26	−11.74 **	−0.69	−2.13	0.48	0.20	7.90 **
**Work engagement (DV), Total effect (c) = 1.12 ****								
Reappraisal	0.29 *	0.78 **	0.89 **	0.23 **	0.04	0.44	0.46	26.57 **
Acceptance	1.17 **	0.35 **	0.71 **	0.41 **	0.20	0.71	0.29	12.71 **
Catastrophizing	−0.81 **	−0.35 **	0.84 **	0.28 **	0.10	0.51	0.27	11.81 **
Rumination	−0.55 **	0.01	1.12 **	−0.01	−0.13	0.10	0.19	7.58 **

Note: age, seniority, and marital status as covariates. *p* < 0.05 *, *p* < 0.01 **.

## Data Availability

The data presented in this study are openly available in https://osf.io/67ne4/?view_only=765250ff510748ae93bc216d622f4572 (accessed on 19 December 2021).

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
