# Peer review of "Preschool Teachers’ Psychological Distress and Work Engagement during COVID-19 Outbreak: The Protective Role of Mindfulness and Emotion Regulation"

_ijerph, 2022, doi:10.3390/ijerph19052645_

Round 1
Reviewer 1 Report
Thanks for the interesting manuscript. A few suggestions:
- It will enhance the readability if the current long introduction can be split into two, e.g., the literature itself can be a separate section;
- It will be great to strengthen the methodology section, e.g., the research design is unclear without any theoretical/conceptual framework and the lack of data analysis framework;
- It will be beneficial to highlight the positive impacts of the proposed mindfulness for teachers in the discussion section;
- It will be useful to include the possible future work especially this is the third year of global pandemic, e.g., what's next?
- It will be necessary to reflect the results to the research questions as this is missing in this manuscript.
Look forward to the revised version. Thanks :)
Reviewer 2 Report
Dear authors, here you can find some suggestions for your manuscript. Best regards.
This paper wants to explore the benefits of mindfulness intervention on the mental health of preschool teacher during the Covid 19. Here you can find some suggestions for your manuscript:
ABSTRACT
L15: it would be useful for the reader to identify the acronym C2C-IT and explain a little bit more about it.
INTRODUCTION
L35: the verb “have” is in present form while “change” is in past, what would be correct?
L37: it would be useful to explain about the normal stress and burnout levels of teachers before explaining of COVID situation, they are in “danger zone” even without a pandemic situation…
L44: this sentence seems to be an opinion, could you please change it in order to seem more scientist?
L93: compassion is considered a different construct, so if you want to speak about it as well, a short definition would help the reader to understand it, because budhist concept of compassion is very different than christian one…
L108: would it be possible that you include a figure with the concepts of emotional regulation? It would be easier for the reader
L118: when talking about the ways that mindfulness can help to achieve more adaptative ER you could make a better paragraph structure, it would be more clear.
L140: if your intervention is about mindfulness and compassion the tittle and introduction should be adapted, because it’s not only a MBI but a compassion intervention as well…in fact, in the way that you describe it I understand that compassion is the key component..
On the other hand, I need more information about the intervention, maybe you should explain it in methods section. A schedule with the sessions content would be very interesting.
L199: did you have inclusion and exclusion criteria?
L224: I would change the Demographic table to the results section
L227: is it a validated questionnaire? If so, please reference it, in the original language and in the language of the study
RESULTS
L290: In table 2 it would be necessary to write in a different way (smaller, indentated…) the levels of stress, anxiety and so on for a better reading of the table.
The results of the table should be explained in a more clear way, in a better structured paragraphs.
DISCUSSION
L332: it is useful to start the discussion talking about the aims of the study and the hypothesis and then commenting the main results and discussing them.
L361: I don’t understand why you are talking about the density of classes here, is it about the stress of teachers? It is not clear
L363: why are you talking about empower and resilience of teachers in this part of discussion? They are not your key concepts
A structure of the discussion depending on the results (ie stress, ER…) would be very useful for the reader
You don’t talk about the compassion components, did you evaluate them? Did you find any changes?
Overall, the manuscript need a better organization, in order for the reader to understand all the concepts you are talking about and follow a clear line in the redaction.
Thank you for considering these suggestions, good luck!
Round 2
Reviewer 2 Report
Dear authors, thank you for providing a new version of your manuscript. Congratulations, you have done a hard work on it, now it has really improved!